# Efficacy Estimation of Microbubble-Assisted Local Sonothrombolysis Using a Catheter with a Series of Miniature Transducers

**DOI:** 10.3390/mi12060612

**Published:** 2021-05-26

**Authors:** Peiyang Li, Wenchang Huang, Jie Xu, Weiwei Shao, Yaoyao Cui

**Affiliations:** 1Academy for Engineering & Technology, Fudan University, Shanghai 200240, China; lipy@sibet.ac.cn (P.L.); xuj@sibet.ac.cn (J.X.); 2Suzhou Institute of Biomedical Engineering and Technology, Chinese Academy of Sciences, Suzhou 215163, China; hwc534@mail.ustc.edu.cn (W.H.); shaowei@sibet.ac.cn (W.S.); 3Division of Life Sciences and Medical, School of Biomedical Engineering(Suzhou), University of Science and Techenology of China, Hefei 230026, China

**Keywords:** deep vein thrombosis, catheter-directed thrombolysis, ultrasound, microbubble, thrombolysis, sonothrombolysis

## Abstract

Intravascular ultrasound has good prospects for clinical applications in sonothrombolysis. The catheter-based side-looking intravascular ultrasound thrombolysis (e.g., Ekosonic catheters) used in clinical studies has a high frequency (2 MHz). The lower-frequency ultrasound requires a larger-diameter transducer. In our study, we designed and manufactured a small ultrasound-based prototype catheter that can emit a lower frequency ultrasound (1.1 MHz). In order to evaluate the safety and efficacy of local low-frequency ultrasound-enhanced thrombolysis, a microbubble (MB) was introduced to augment thrombolysis effect of locally delivered low-intensity ultrasound. The results demonstrated that combination of ultrasound and MB realized higher clot lysis than urokinase-only treatment (17.0% ± 1.2% vs. 14.9% ± 2.7%) under optimal ultrasound settings of 1.1 MHz, 0.414 MPa, 4.89 W/cm^2^, 5% duty cycle and MB concentration of 60 μg/mL. When urokinase was added, the fibrinolysis accelerated by MB and ultrasound resulted in a further increased thrombolysis rate that was more than two times than that of urokinase alone (36.7% ± 5.5% vs. 14.9% ± 2.7%). However, a great quantity of ultrasound energy was required to achieve substantial clot lysis without MB, leading to the situation that temperature accumulated inside the clot became harmful. We suggest that MB-assisted local sonothrombolysis be considered as adjuvant therapy of thrombolytic agents.

## 1. Introduction

Deep vein thrombosis (DVT) refers to the abnormal formation of thrombi in the deep venous system, which often occurs in the lower extremities [1]. It is estimated that the annual incidence is around 5 per 10,000 among the general population [2], whereas in some areas of Norway and America, the figure can be up to 9.3 per 10,000 in people aged 20 years and older [3] and 11.7 per 10,000 in people over 45 years old [4]. When the thrombus of DVT falls off from the site of formation and travels with the reflux of venous blood, it can lead to a potentially serious blockage in the pulmonary artery, termed pulmonary embolism (PE), which is a life-threatening complication of DVT [5]. Post-thrombotic syndrome (PTS) is another common, chronic complication of DVT. Its clinical manifestations range from leg pain, swelling to severe symptoms such as intractable oedema and leg ulcers [6]. It is reported that approximately 15–50% of patients with an initial DVT develop PTS [7]. PTS significantly impairs health-related quality of life among patients [8,9], and constitutes a heavy economic burden to both families and the healthcare system [10,11].

Therefore, preventive measures are of great importance for patients with DVT to lower the possibility of extending to acute and chronic complications. Although the standard therapy for venous thromboembolic disease recommended by the American College of Chest Physicians is anticoagulant [12,13], it can hardly remove clots by itself or minimize the development of PTS if sufficiently [14]. Recent reviews have shown that catheter-directed thrombolysis (CDT) therapy combined with anticoagulant, has more advantages over anticoagulant systematic treatment in improving the coagulation and decreasing the likelihood of PTS development [15,16]. CDT benefits from effective administration of medication into the thrombus directly, so as to increase the concentration of medication within the clot, and decrease the amount of thrombolytic agent which is often linked to increased bleeding risk. 

In recent years, ultrasound-assisted CDT has emerged as a promising method for the thrombotic disease. Clinical trials reported that ultrasound (US) in conjunction with regional thrombolysis could reduce complication rates in the treatment of venous occlusive diseases [17]. Ultrasound-assisted CDT also performed well in reversing right ventricular dysfunction and dilatation in patients with intermediate-risk PE, while existed as a potentially promising technique for high-risk PE [18,19]. To the best of our knowledge, currently, the only commercially available intravascular thrombolysis equipment is EkoSonic Endovascular System (EKOS Corporation, Bothell, WA, USA), which is designed to amplify the thrombolysis efficacy of CDT through high-frequency, low-intensity ultrasound [20]. The safety and efficacy for recanalization of acute ischemic stroke and delayed treatment of lower extremity DVT in clinical applications have been confirmed [21,22]. The possible mechanisms of the thrombolysis enhancement are attributed to the increased transport of enzymes into the clot during ultrasound exposure [23]. More specifically, noncavitational effects of ultrasound was believed to play a role in the fibrinolysis process [24]. However, intravascular ultrasound-assisted CDT suffers from long treat times (mean infusion time of 22.4 h) and highly depends on the dose of the thrombolytic agent [17]. It was demonstrated that the regional ultrasound energy is insufficient to produce substantial clot weight reduction in the absence of thrombolytic drugs in vitro [24]. Since Tachibana and Tachibana pioneered the idea of accelerating thrombolysis by combining contrast agent microbubbles (MB) with ultrasound, cavitation activity [25] and radiation forces have been demonstrated to be likely responsible for this acceleration in an in vitro model [26,27]. 

We assumed that local ultrasound-enhanced thrombolysis assisted by MB will facilitate clot lysis due to the decreased cavitation threshold. However, the acoustic and non-acoustic parameters involved are poorly understood.

Although many studies have been conducted on thrombolysis with focused ultrasound in vitro, its clinical value is poor due to its larger size. The forward-looking intravascular ultrasound may enable the generation of higher pressures, but it is not suitable for blood clots in longer areas of the vessel. It has been comprehensively demonstrated that low-frequency ultrasound can enhance the lytic rate [28]. The larger-diameter transducer is required for the lower-frequency ultrasound. In our work, the design of the side-looking intravascular transducer realized low-frequency ultrasound-enhanced thrombolytic efficiency. Compared with the Ekosonic catheters (2 MHz) used clinically, we can emit a lower frequency ultrasound (1.1 MHz) with a small diameter of the ultrasound-based prototype catheter. With lower transmitting power, we didn’t need the cooling device. Our study will have a good prospect of clinical application in the future.

The purpose of this study was to evaluate the safety and efficacy of local ultrasound-enhanced thrombolysis in three ultrasound regimes (US alone, US + MB, US + MB + urokinase). All bovine blood clots received sonification from an internal ultrasound produced from a manufactured catheter system. Parameters related to clot lysis rate were explored in US + MB experiments, including MB concentration, input voltage, duty cycle and treatment times. Thrombolysis efficacy was assessed by absolute clot lysis rate compared with that in the control group.

## 2. Materials and Methods

### 2.1. Subsection Preparation of Bovine Blood Clots

According to a previous method [29], the bovine blood clots were prepared by mixing the whole bovine blood (Zhengzhou Jiulong Biological Products Co., Ltd., Zhengzhou, China) with 0.5 M (mol/L) calcium chloride solution in a polypropylene centrifuge tube (2 mL, diameter of 10.2 mm, length of 42.5 mm). Then, the tube was sealed and incubated in a 37 °C water bath for 3 h. After that, the clot models were basically formed, and stored at 5 °C to maximize clot retraction and resistance to lysis [30]. In order to explore optimal parameters of the stable bovine blood clots, we used the control variable method, taking the volume ratio of anticoagulant bovine whole blood to calcium chloride and the cold preservation time as variables. Then the clot was placed in static normal saline to observe the stability over time, which is called the immersion method. Considering the dissolution of bovine red blood cells and separation of serum in static saline, we assumed that clot lysis can increase with standing time. If the reduction rate of thrombus soaked in static normal saline had little difference within 10~90 min, we believe that the in vitro clots prepared by these parameters are the most stable. We fixed the volume of bovine blood at 1 mL, the volume of coagulator was set as 10, 15, 20, 25, 30, 50, 60, 70, 80, 90, 100, 150, 200, 250, 300 μL, while the cold preservation time was controlled at 24 h. The optimal volume ratio of the clots was found by the immersion method. Then, fixed the optimal volume ratio of clot preparation (1 mL bovine blood with 70 uL calcium chloride solution), the cold preservation time was set as 0, 24, 48, 72, 96, 120, and 144 h. The optimal cold preservation time was found by the immersion method. The total number of bovine blood clots used in the following thrombolysis experiments was 96.

### 2.2. Ultrasound-Based Prototype Catheter and Experimental Platform

The custom-made piezoelectric ultrasonic transducer (Suzhou GuoKe Ultra Medical Technology Co., Ltd.,Suzhou, China) had a miniature size of 2 mm length, 0.4 mm width and 0.35 mm thickness. The physic properties and materials of the ultrasound-based prototype catheter are listed in Table 1. Referring to Figure 1, the ultrasound core wire was manufactured by mounting two rows of the transducers (2/row) evenly on the opposite side of the conductive sheet, the sheet served as a common cathode. The transducers, spaced at 5 mm from each other unilaterally, consequently generated a superimposed uniform unfocused acoustic field around the central axis. The ultrasound-based prototype catheter is shown in Figure 2, to prolong the life-span of the catheter, the ultrasound-based prototype catheter includes a fluorinated ethylene propylene (FEP) sleeve (1.1 mm ID, 0.2 mm wall thickness) that houses the ultrasound core wire and a miniature K-type thermocouple connected to the transducer surface. The K-type thermocouple was applied to measure the temperature of the thrombus interior. The spatial-peak pressure amplitude was 0.267 MPa at a distance of 6 mm from the surface of the transducer without an FEP sleeve with a needle hydrophone in the water tank. Under the same conditions, when the FEP sleeve housed the transducer, the spatial-peak pressure amplitude was 0.253 MPa. In the presence of the FEP sleeve, the spatial-peak pressure amplitude was 5% less than the pressure at the transducer surface. Because the beam was going to diverge, attenuation was found to be acceptable. The thrombolysis catheter system is designated to describe this kind of fabrication as shown in Figure 3, with a maximal lateral dimension of about 2.5 mm. In order to achieve effective administration of medication into the treatment zone, a drug-delivery catheter (outer diameter of 0.9 mm) integrated with the ultrasound-based prototype catheter was modified by sealing the outlet and pricking multi-side holes.

The schematic of the thrombolysis experimental system is illustrated in Figure 4. The ultrasound transducers were driven by sinusoidal bursts generated from an arbitrary wave generator (33600A, Agilent Technologies, Inc., Santa Clara, CA, USA) and then amplified by a radio frequency amplifier (150A100B, AR, Inc., Bothell, WA, USA). Meanwhile, the drug-delivery tube was connected to a syringe pump (Xinka Electronic Products Co., Ltd., Beijing, China). Water temperature in the tank was maintained constant at 35.7 °C.

### 2.3. Test of Acoustic Characteristics 

Since the ultrasound transducer was designed for the cavitation effect, the selection of appropriate frequency was of significance before the experiment. The impedance curve of the ultrasonic transducer was measured by the impedance analyzer (E4991A, Agilent Technologies, Inc., Santa Clara, CA, USA). It was used to determine the optimal resonance frequency. Acoustic characteristics of the transducer were tested in a larger water tank filled with deionized, degassed water under the same setup as the experiment. A sinusoidal burst signal with 10 cycles and 500 Hz pulse repetition frequency (PRF) was applied to drive the transducers. Acoustic pressure output was calculated by conversion of raw voltage output received from a needle hydrophone (NH0500, Precision Acoustics, Ltd., Dorchester, Dorset, UK). Spatial peak pulse average intensity is defined as pulse intensity integral divided by pulse duration, while spatial peak temporal average intensity is also derived from the pulse intensity integral but multiplied by PRF. The mechanical index (MI) is defined as peak rarefactional acoustic pressure (in MPa) divided by the square root of the operating frequency (in MHz) without consideration of ultrasonic attenuation of tissue. The hydrophone was aligned vertically to the surface of each transducer on a precision positioning stage (TS303, BOCIC, Ltd., Beijing, China) at a distance of 2 mm.

### 2.4. In Vitro Experiment Protocol

The thrombolysis efficacy was evaluated by absolute clot lysis rate (ALR, %) and absolute velocity of clot lysis (AVLR, %/min) which almost neglect clot mass reduction bias due to the non-ultrasound effect. ALR (%) is defined as the difference between the percentage of mass reduction in the active group and the percentage of mass reduction in the control group. AVLR (%/min) is defined as the ratio of ALR to the corresponding time. The expressions are as follows,
(1)$$ALR=\left(\frac{w_{0}-w_{1}}{w_{0}}-\frac{w_{c0}-w_{c1}}{w_{c0}}\right)\times 100\%$$
(2)$$AVLR=\frac{ALR}{\Delta T}.$$

Here, $w_{0}$ is the initial clot mass and $w_{1}$ is the residual clot mass. Subscripts with c indicate data from the control group. ΔT is the time duration of ultrasound exposure.

For all experiments, the bovine blood clots were transferred to new centrifuge tubes after being weighed on an electronic balance (AR323CN, OHAUS, Inc., Souderton, PHL, USA). The average initial weight of clot samples was 925 mg ± 59 mg. Then the catheter system was gently inserted into the middle of the clot through the outlet of the sample tube. The tube was completely immersed in 0.9% saline solution (35.7 °C). Both the sample tube and catheter system were kept in an immobile position. Ultrasound exposure persisted throughout the whole thrombolysis process. After 60 min insolation, the catheter system inserted into the thrombus was gently pulled out. The residual clots were flushed with 0.9% saline solution and dried on filter paper to measure the final mass. All thrombolysis experiments were repeated 3 times for calculation of the mean and standard deviation. In order to avoid clot lysis arose from non-ultrasound effects, such as mechanical damage caused by the catheter inserting process, spontaneous degradation and medication infusion, the control group tests were performed following the same procedures as previously described. However, ultrasound exposure was left out, and an equivalent quantity of 0.9% saline served as a placebo in the control. For urokinase alone treatment, the clots were dissolved with 1 mL urokinase solution (20,000 units/mL) in 60 min using a catheter system. All clot models in active groups (US alone, US + MB and US + MB + urokinase) were insonified by pulsed ultrasound. The ability of ultrasound alone to dissolve thrombus has been studied by changing the duty cycle between 10% and 50% in 10% increment preliminarily. The duty cycle was varied by changing the burst counts for a given PRF of 500 Hz. The influence of MB concentration, input voltage, duty cycle and time duration of ultrasound were analyzed in the US + MB experiment. To further increase clot lysis, urokinase with various concentrations was applied in MB-assisted ultrasound-enhanced thrombolysis experiments.

The MB (SonoVue, 59 mg/bottle, Bracco Suisse SA) was prepared according to the manufacturer’s instructions by injecting 0.9% saline solution into a SonoVue vial and then shaking the whole system for 20 s to generate milky-white suspension. The different sizes of reconstituted MBs, ranging from 1 μm to 10 μm, have a resonance frequency spectrum between 1 MHz and 10 MHz [31]. The typical concentration is 2~5 × 10^8^ bubbles/mL or 45 μg/mL (SF_6_) when 5 mL saline solution was injected. We shook the vial to redistribute MBs homogeneously before each experiment, and the microbubble injecting rate was 1 mL/h. Thrombolytic agent urokinase (manufacturer Tianjin Biochem Pharmaceutical Co., Ltd., Tianjin, China) has a specification of 100,000 units/piece. It was diluted with 5 mL 0.9% saline solution prior to use to form urokinase solution with 20,000 units/mL concentration. Variable concentration was realized with a variable volume of saline in the same way. 

Statistical analysis was executed using SPSS20.0 statistical software package. Data of mass reduction percentage were analyzed by one-way analysis of variance. The test of homogeneity of variance was performed ahead of analysis to determine whether there was the necessity to do data transformation, of which logarithmic transformation was a preferred choice. Statistical significance was set at *p* < 0.05.

## 3. Results

### 3.1. Determination of Clot Preparation Protocol

Calcium ion acted as coagulant was studied to determine the applicable dosage as shown in Figure 5a. The experiments were conducted in 60 min with bovine blood clots free from cold preservation treatment. It can be seen that CaCl_2_ can be used as a coagulant to obtain a relatively stable clot to a large extent of volume, at least from 50 μL to 200 μL. Once the volume exceeded 200 μL, clot lysis increased rapidly. Clot models prepared with 300 μL CaCl_2_ showed a great statistical difference in clot lytic rate than any other treatment (*p* < 0.001). The time of cold preservation is another factor that affects the stability of the formed thrombus. Clot lysis in clot models prepared from 70 μL calcium chloride was observed. As we can see in Figure 5b, the clot lytic rate surpassed 20% only in experiments that lacked cold storage. No significant difference (*p* > 0.05) was shown among samples preserved for more than 24 h. In static saline solution, the thrombus showed robust stability independent of standing time (Figure 5c). No significant clot lysis difference was observed during the experiment. Therefore, all bovine blood clots used in later experiments were stored in cold temperature for at least 24 h prior to use, and 70 μL CaCl_2_ was selected to promote thrombosis. Since clot lysis does not increase with standing time, we suppose results from control groups of 60 min can be also used to compare with active groups of various duration.

### 3.2. Acoustic Characteristics of the Transducer

Considering that decreased cavitation threshold or higher thrombolysis efficiency can be realized at a lower frequency [28,32], the experimental resonance frequency of the transducer was selected to be 1.1 MHz according to the impedance analysis result (Figure 6a). The blue curve stands for the impedance vs. frequency, and the orange curve means phase angle vs. frequency. The peak negative pressure (PNP) conversion profile is shown in Figure 6b for an input voltage, frequency, PRF and burst counts of 30–70 V_pp_, 1.1 MHz, 500 Hz and 10, respectively. The measured PNP of ultrasound transducers ranges from 0.19 MPa to 0.48 MPa, on average, depending on the input voltage of excitation. The experiment result showed that there is a good linear correlation between PNP and peak to peak voltage of input. Thus, the linear fitting was applied and the equations were listed below. The typical time-domain waveform of measured acoustic pressure is shown in Figure 6c (at 1.1 MHz, 60 V_pp_, 10 bursts). The parameters of acoustic intensity and mechanical index were summarized in Table 2 under the condition of 1.1 MHz and 60 V_pp_. Indeed, the acoustic pressure and intensity would be larger than the current values on the surface of each transducer.

The distribution of the sound field perpendicular to the center axis of the transducer was measured by the hydrophone and the three-dimensional precision acoustic field scanning system (UMS3.0 Precision Acoustics Ltd. Dorchester, Dorset, UK), and the scanning was started from a position 1.7 mm from the surface of the transducer. The relationship between voltage squared integral and distance is shown in Figure 7. The maximum point of sound pressure is 2 mm away from the surface of the transducer. A diagram of the plane acoustic field scanning 2 mm away from the surface of the transducer is shown in Figure 8, the *Y*-axis is along the axial direction of the catheter axis, and the *X*-axis is along the lateral direction of the catheter axis. There was no wave interruption by placing the wire in the middle of the matching found in the sound field distribution results. In addition, the wavelength of the 1.1 MHz ultrasonic waves in water is nearly 1363 μm which is much larger than the core wire diameter of 40 μm, so the effect of core wires on the propagation of sound waves can be ignored. The sound pressure distribution is humped, and the sound pressure reaches a peak in front of the two transducers. The sound pressure is not symmetrical, which may be caused by the error of the adhesion of the transducer.

### 3.3. In Vitro Experiments 

Ultrasound-enhanced thrombolysis experiments were performed in in vitro bovine blood clots. The clot mass reduction rate of the control group for US alone, US + MB and US + MB + urokinase is 28.2% ± 3.6%, 30.7% ± 1.2%, and 32.8% ± 1.8%, respectively. The only difference among the control groups is the dose of infused placebo (0 vs. 1 mL vs. 2 mL). The clot lytic rate of urokinase alone therapy is 45.6% ± 2.7%, among which only 14.9% reflects the authentic fibrin degradation efficacy of urokinase (20,000 units/mL) after ignoring the clot mass loss resulted from operation and infusion (30.7% ± 1.2%).

With the input condition of 1.1 MHz, 50 V_pp_ and 500 Hz PRF, despite the US alone group showing that there was an enhancement in thrombolysis rate with an increasing duty cycle (Figure 9), we have to concern about the phenomena of thrombus burning and increased adhesiveness to the transducer, which occurred as a result of duty cycle more than 30%. In addition, almost no mass reduction (−0.1% ± 1.0%) occurred in clot models exposed to 10% duty cycle ultrasound. We then conducted further tests to figure out the thermal mechanism inside the clot. The input voltage was increased from 50 V_pp_ to 60 V_pp_, and the temperature was collected after US exposure durations for 5 min. Figure 10 provides the fact that as the duty cycle increased to 50%, the temperature increased by almost 6 °C. The rising tendency observed was confined within 5 min. Since temperature rise may cause thermal damage to the vascular wall in clinical practice, no more than a 10% duty cycle was allowed in the following experiments. Temperature rise (0.9 °C) observed at 10% duty cycle was negligible.

For MB-assisted ultrasound-enhanced thrombolysis, the initial in vitro test (Figure 11) was conducted using a typical MB concentration of 45 μg/mL and sinusoidal pulse excitation of 60 V_pp_ and 5% duty cycle. After 60-min ultrasound exposure, the mass of target clots decreased by 44.4% ± 1.2%, and the ALR and AVLR were measured 13.6% and 0.23%/min, which is comparable to urokinase alone (14.9%, 0.25%/min). Figure 12a illustrates that clot lysis rate increased with the increase of MB concentration, which is consistent with previous studies [33,34]. Remarkably, clot lysis rate exceeded that of urokinase alone when MB concentration greater than 60 μg/mL. The influence of ultrasound parameters on MB-assisted sonothrombolysis efficacy is demonstrated in Figure 12b–d. Figure 12b illustrates that a greater clot lysis rate resulted from higher input voltage with the exception of 70 V_pp_, which showed nearly equal efficiency with 60 V_pp_. Similarly, Figure 12c indicates clot lysis rate grew with the increasing duty cycle except for 5% and over. Note that an extremely significant difference was shown between 0.5% and 1% (*p* < 0.001), and substantial enhancement of clot lysis (12.7%) was achieved in relatively low ultrasound energy (1% duty cycle). Figure 12d shows that the clot lysis rate increased as time went by. The same result as urokinase alone could be achieved in 40 min under the condition of 60 V_pp_, 5% ultrasound pulse excitation and 60 μg/mL MB concentration (14.4% ± 0.3% vs. 14.9% ± 1.2%). In terms of the absolute velocity of clot lysis, the first 10 min (0.7%/min) is almost triple of other 10-min time duration.

Finally, Figure 13 illustrates that further clot lysis rate was achieved when urokinase was added into MB-assisted sonothrombolysis. The highest absolute percentage of clot mass reduction (36.7% ±5.5%) was present when urokinase of the highest concentration (20,000 units/mL) was administered simultaneously, which is more than twice that of urokinase alone.

## 4. Discussion

We were intended to evaluate the efficacy of local ultrasound-enhanced thrombolysis, and explore the optimal parameters for MB-assisted sonothrombolysis in in vitro bovine blood clots. 

The stability of the bovine blood clot was connected with two possible factors, including the volume ratio of bovine blood and 0.5 M (mol/L) CaCl_2_ and the cold storage times for clot retraction. It was found that only a limited range of the volume ratio (1:50 to 1:200, mL:μL) could potentiate blood clot formation. Once the calcium ion is excessive, anticoagulation would dominate in the thrombosis process. The coagulated clots revealed less effective mass loss percentage when stored in low temperature, which is likely associated with increased density of fibrin network [35]. However, no significant difference was shown among samples stored for various duration. That may inspire us to prepare clot models in lower formation times instead of more than three days before experiments [26,29,33,36]. In addition, regarding to clot formation times or thrombus age [37], a more effective thrombolysis effect was expected to realize in fresher thrombus which enables us to gain an edge in the treatment of acute DVT.

The custom ultrasonic transducer was demonstrated to have a good linear relationship between output acoustic pressure (PNP in MPa) and input voltage (in V_pp_). Hence, the cavitation threshold can be easily accessed by changing the input voltage. For 60 V_pp_, the most commonly used input voltage parameter in MB-mediated thrombolysis, the measured PNP (304 kPa in average), is slightly above the inertial cavitation threshold (250 kPa) provided by Petit et al. [27], which measured at a similar operation frequency of 1 MHz. The corresponding spatial peak pulse average intensity (4.89 W/cm^2^) surpasses the power (0.5 W/cm^2^) suggested by the standard EkoSonic MACH4e protocol for some clinical studies [38,39]. It is worth noting that greater acoustic parameters would be measured on the surface of the transducer, so both stable cavitation and inertial cavitation were likely present in the ultrasound insonation zone.

We have demonstrated that MB-assisted local sonothrombolysis can realize an equal and even higher absolute thrombolytic rate compared with urokinase alone protocol. The typical clot mass reduction of 17.0% was observed at ultrasound settings of 1.1 MHz, 0.414 MPa, 4.89 W/cm^2^ (corresponding to 60 V_pp_), 5% duty cycle, 500 Hz PRF and MB concentration of 60 μg/mL. However, without the presence of exogenous cavitation nuclei, the MB existing in the clot was not sufficient for locally delivered acoustic energy to produce substantial clot lysis (Figure 9). Even though the acoustic power applied in US alone is fairly lower than high intensity focused ultrasound (100 to 10,000 W/cm^2^), which stems from the medical application of ultrasound thermal effect [40], ultrasound transducer clinging to clot still possibly incurs temperature rise internally. During our experiment, thermal mechanisms are not believed to take part in the clot lysis process, because the temperature rise involved in MB-mediated thrombolysis was negligible. A previous study using Ekosonic catheter has also excluded the heat effect in ultrasound-accelerated rt-PA-induced thrombolysis [24]. Pulsed ultrasound potentially kept a balance between temperature accumulation and thermal dissipation inside the clot when duty cycle less than 5%, thus the saline coolant used in EKOS catheter systems [17,41] may not be a requirement for MB-assisted low-intensity sonothrombolysis. The commercial endovascular thrombolysis catheters are designed by utilizing the benefits of ultrasound to facilitate the transport of fibrinolytic enzymes into the clot. Without the adjuvant thrombolytic agent, the energy of US alone is insufficient to result in obvious clot weight loss [24]. Our results suggest MB be used as an alternative thrombolytic enhancer in combination with a relative lower-frequency (1.1 MHz vs. 2.2 MHz in EkoSonic endovascular system) ultrasound. The cavitation threshold of 1.1-MHz ultrasound seemed to be PNP of 0.3361 MPa when 50 V_pp_ resulted in more significant clot lysis than 40 V_pp_ (11.6% vs. 4.0%, *p* < 0.01). A higher clot lysis rate can be realized under the condition of higher MB concentration, input voltage (corresponding to PNP), duty cycle (corresponding to acoustic power) and longer treatment times. Interestingly, unlike conclusions from previous researches that thrombolysis efficacy increased proportionally to acoustic parameters [28,33], no more dramatically higher absolute clot lysis was observed in group 70 V_pp_ (Figure 12b), group 7% (Figure 12c) and 9% (Figure 12c) in our study. This may result from the thorough red blood cell loss that occurred in the limited cavitation zone, therefore the ultrasound exposure should be moved to an area closer to the undissolved clot. Ultrasound dissolved part of the clot when the duty cycle was higher than 5% and the voltage was higher than 60 Vpp. Because there is no dynamic flow, the dissolved clot may stick to the surface of the catheter, preventing subsequent transmission of the ultrasound. With the increase of duty cycle and voltage, the biological effect of ultrasound may be mainly thermal effect. The higher temperature may change the physical properties of the transducers and the clot, which may be one of the reasons why there was no more increase in the lysis rate. The highest clot lysis velocity appeared in the first 10 min (0.7%/min), it may due to massive clot weight loss that occurred during the transfer handling. It is also possibly associated with the inhomogeneous distribution of MB in the infusion pump, as the accumulation of MB in the upper level of suspension was observed during the experiment. With the optimal acoustic parameters (1.1 MHz, 0.414 MPa, 5% duty cycle, 500 Hz PRF) and 60 μg/mL MB concentration, the thrombolytic rate can be augmented from 0.79%/min to 1.16%/min when fibrinolysis of urokinase was accelerated by microstreaming and shear stress-induced from acoustic cavitation. 

The thrombolysis efficacy of the multi-element catheter system is subject to the arrangement of transducers. Compared with forward-looking fabrication with a confined field of insonation [33], the side-looking transducers show their competence in expanding the therapy area by increasing the number of transducers. 

One of the advantages of MB-assisted local sonothrombolysis is that MB infused through multi-side holes maintains high concentration inside the clot. Since the control groups (0, 1 mL/h, 2 mL/h) with faster injection rate revealed higher clot lysis, we suppose that infusion flush can cause clot debris that is a potential factor of distal embolism. Therefore, a slow injection rate is relatively secure in dissolve thrombus. Another advantage is the use of MB less than recommended dose injected for ultrasonic imaging clinically (2.4 mL for vascular enhancement). A limitation of the present study is that the coarse surface of the manufactured catheter system caused fragmented clot particles during the inserting and pulling out process. Thus, a smoother coating is needed for further catheter development. Although we have demonstrated the safety of thermal effect is warranted in a centrifuge tube, a blood vessel phantom with the same radial size of lower extremity artery is still required for a more precise test.

## 5. Conclusions

In this study, the results are shown that MB plus locally delivered low-intensity pulsed ultrasound can be effective in dissolving clots without any use of a thrombolytic agent. Under the condition of optimal acoustic parameters (1.1 MHz, 0.414 MPa, 4.89 W/cm^2^, 5% duty cycle, 500 Hz PRF), greater thrombolysis efficacy can be realized compared with urokinase alone treatment. Fresher MBs are expected to shorten the treatment. For a clinical emergency, a minor thrombolytic agent was suggested to be used in combination with ultrasound and MB to enhance fibrinolysis. Our future work will investigate the usefulness of an improved catheter system in an in vivo model.

## Figures and Tables

**Figure 1 micromachines-12-00612-f001:**
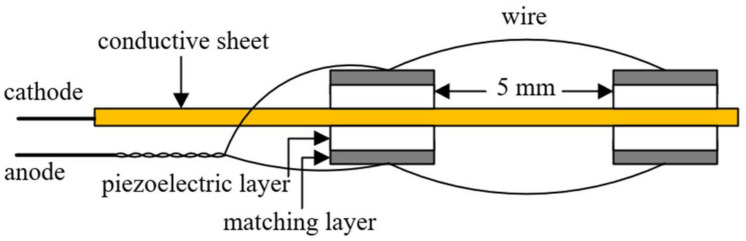
Schematic wiring diagram of the ultrasound core wire.

**Figure 2 micromachines-12-00612-f002:**
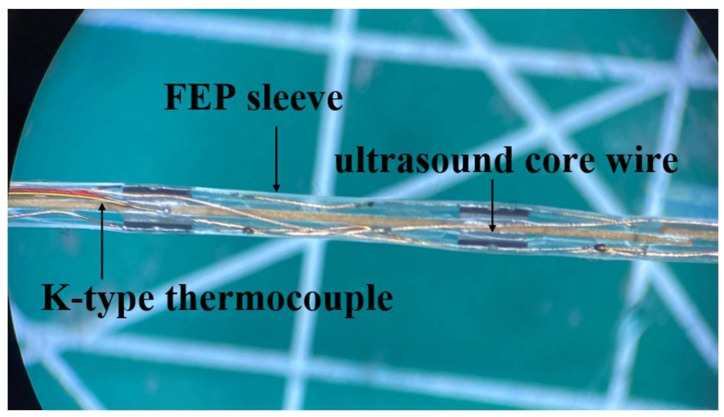
Ultrasound-based prototype catheter.

**Figure 3 micromachines-12-00612-f003:**
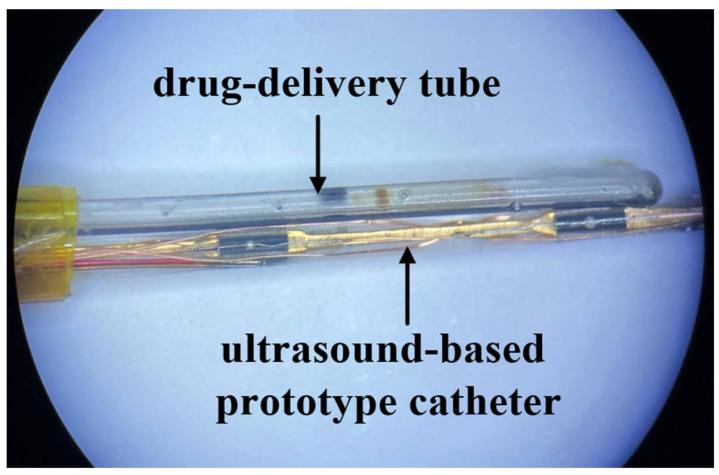
Thrombolysis catheter system.

**Figure 4 micromachines-12-00612-f004:**
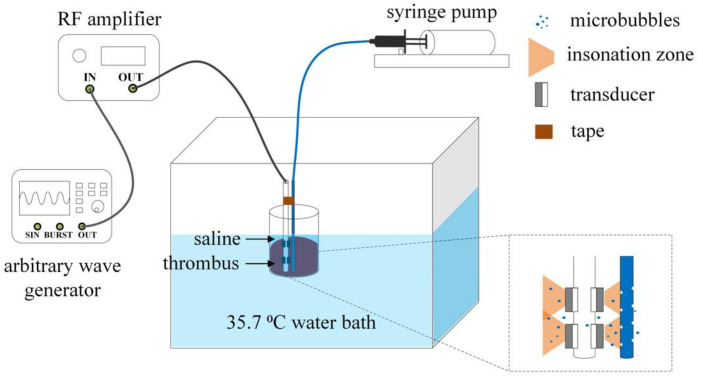
Schematic of thrombolysis experimental system.

**Figure 5 micromachines-12-00612-f005:**
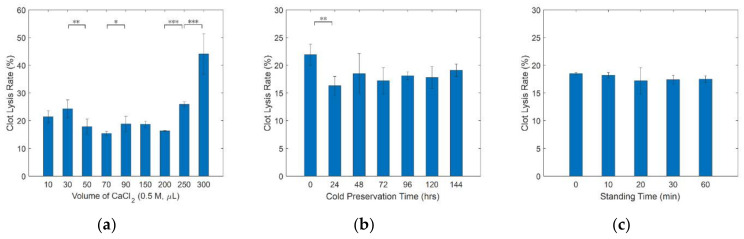
Mass reduction of bovine blood clots in static saline. (**a**) Clot lysis rate of clots prepared at each volume of CaCl_2_. Clots were exposed in saline for 60 min without storing at 5 °C in advance (*n* = 3). (**b**) Clot lysis rate for various cold preservation times. Clots formed with 70 μL CaCl_2_ were exposed in saline for 60 min (*n* = 3). (**c**) Clot lysis rate over different standing times. Clots were prepared with 70 μL CaCl_2_ and stored in 5 °C for at least 24 h before experiment (*n* = 3, * *p* < 0.05, ** *p* < 0.01, *** *p* < 0.001).

**Figure 6 micromachines-12-00612-f006:**
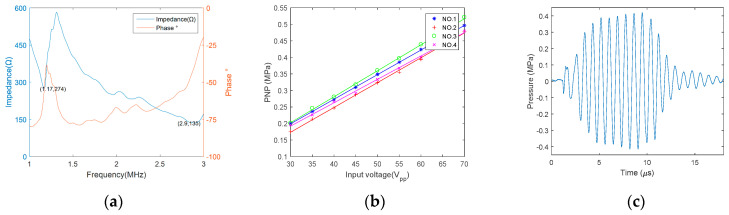
(**a**) Impedance amplitude and phase curves of ultrasound transducer. (**b**) Peak negative pressure conversion profile of transducers in one ultrasound-based prototype catheter. Distance of each transducer to the hydrophone is 2 mm. The fitted formula for each transducer is (1) y=0.0074x−0.0243,  R^2^ = 0.9999; (2) y=0.0075x−0.0518, R^2^ = 0.9988; (3) y=0.0079x−0.0334, R^2^ = 0.9992; (4)  y=0.071x−0.0199, R^2^ = 0.9984. (**c**) Time domain waveform of acoustic pressure.

**Figure 7 micromachines-12-00612-f007:**
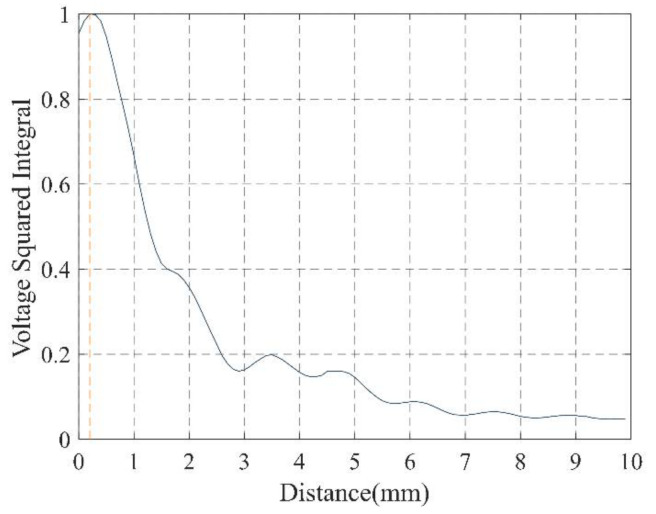
The relation between voltage squared integral and distance.

**Figure 8 micromachines-12-00612-f008:**
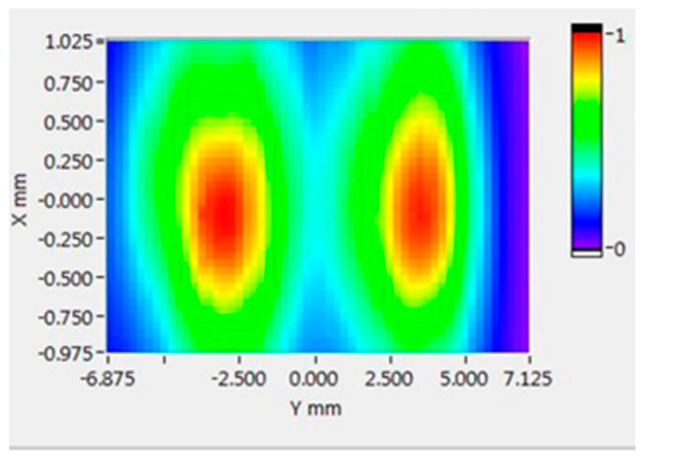
The normalized spatial distribution diagram of the sound field.

**Figure 9 micromachines-12-00612-f009:**
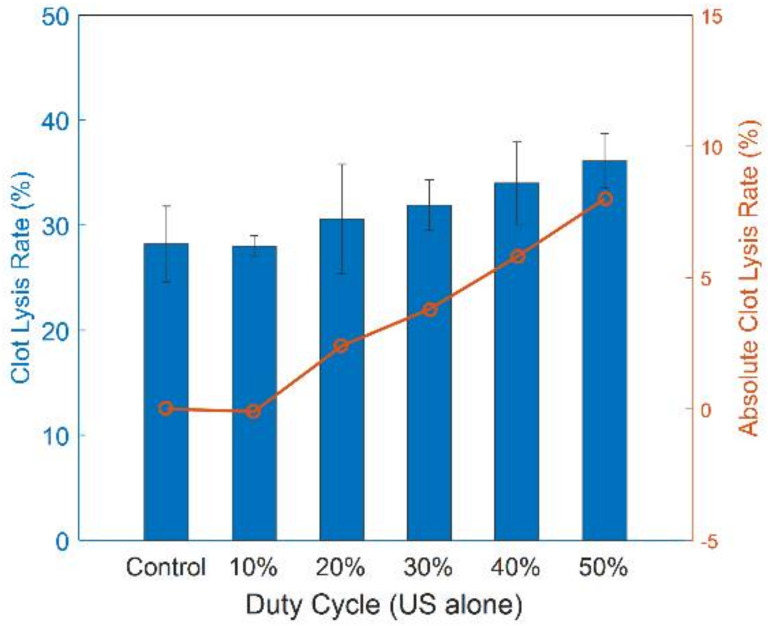
In vitro US alone experiment result. Clot lysis vs. duty cycle (*n* = 3, US settings of 1.1 MHz, 50 V_pp_ input voltage, 500 PRF, 60 min treatment).

**Figure 10 micromachines-12-00612-f010:**
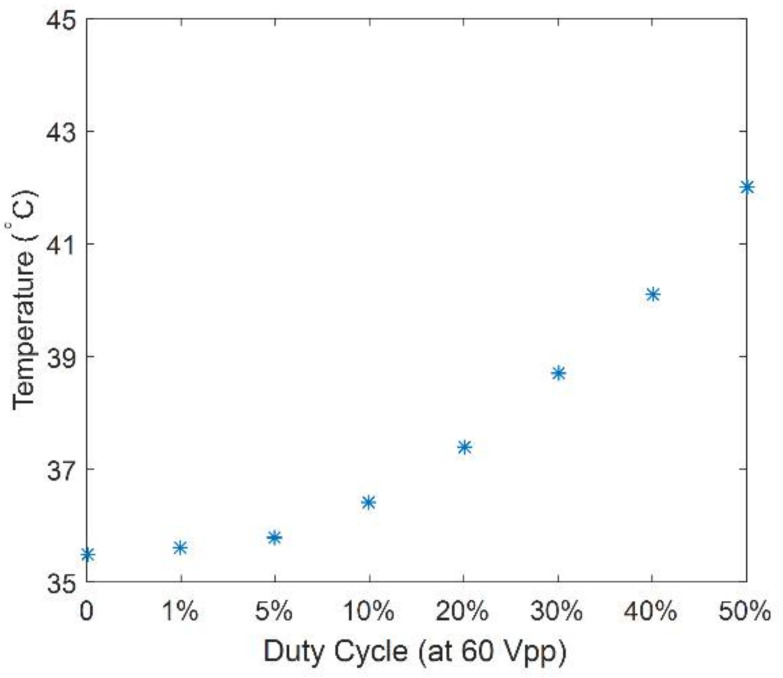
The temperature rise of thrombus interior at each duty cycle (at 1.1 MHz, 60 V_pp_ input voltage, 500 PRF).

**Figure 11 micromachines-12-00612-f011:**
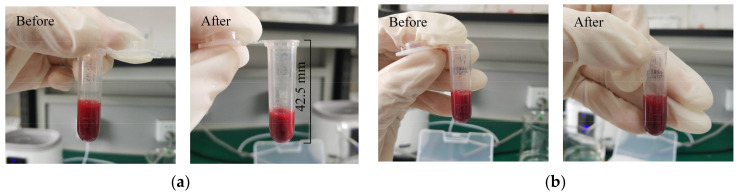
Initial in vitro test for MB-assisted local sonothrombolysis. (**a**) Clot mass reduction in an active group before and after treatment. (**b**) Clot mass reduction in the control group before and after treatment.

**Figure 12 micromachines-12-00612-f012:**
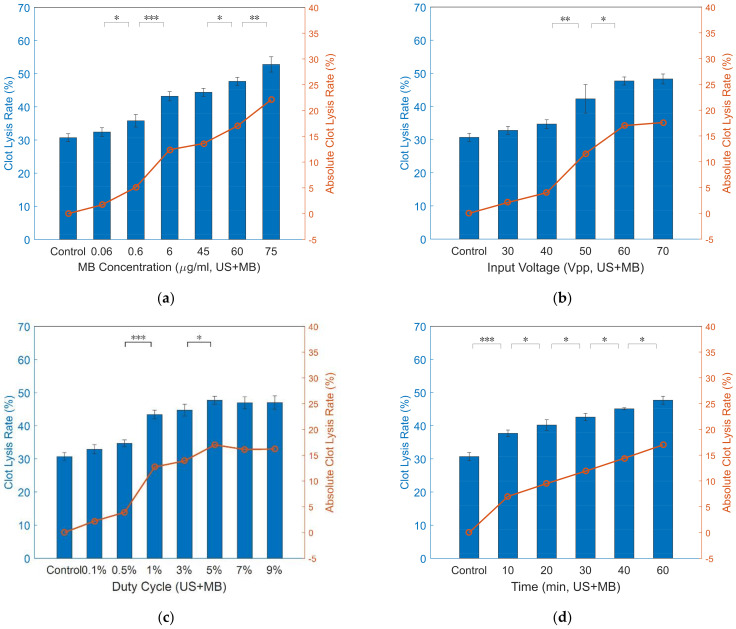
In vitro US + MB experiment results. (**a**) Clot lysis vs. MB concentration (*n* = 3, US settings of 1.1 MHz, 60 V_pp_ input voltage, 500 PRF, 5% duty cycle, 60 min treatment). (**b**) Clot lysis vs. input voltage (*n* = 3, US settings of 1.1 MHz, 500 PRF, 5% duty cycle, 60  μg/mL, MB injected at 1 mL/h, 60 min treatment). (**c**) Clot lysis vs. duty cycle (*n* = 3, US settings of 1.1 MHz, 60 V_pp_ input voltage, 500 PRF, 60  μg/mL, MB injected at 1 mL/h, 60 min treatment). (**d**) Clot lysis vs. time (*n* = 3, US settings of 1.1 MHz, 60 V_pp_ input voltage, 500 Hz PRF, 60  μg/mL, MB injected at 1 mL/h) (*** *p* < 0.001, ** *p* < 0.01, * *p* < 0.05).

**Figure 13 micromachines-12-00612-f013:**
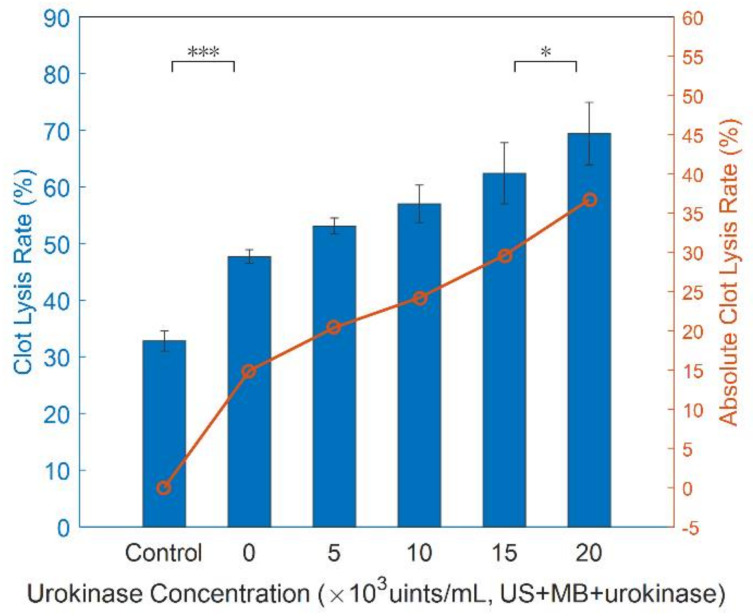
In vitro US + MB + urokinase experiment result. Clot lysis vs. urokinase concentration (*n* = 3, US settings of 1.1 MHz, 60 V_pp_ input voltage, 500 Hz PRF, 60 μg/mL MB in combination with 20,000 units/mL urokinase injected at 2 mL/h, 60 min treatment) (*** *p* < 0.001, * *p* < 0.05).

**Table 1 micromachines-12-00612-t001:** The physic properties of the ultrasound-based prototype catheter.

Composition	Piezo Layer	Matching Layer	Conductive Sheet	Core Wire
Material	PZT4	Conductive Glue Mixture	Copper	Silvre Plated Copper Alloy
Dimension (um)	2000 × 400 × 350	100	100	40
Young’s module (GPa)	63	11	-	-
Poisson’s ratio	0.34	0.3	-	-
Q_m_	500	-	-	-
Density (kg/m^3^)	7550	1350	-	-
Sound speed (m/s)	4800	2200	-	-

**Table 2 micromachines-12-00612-t002:** The averaged values for acoustic parameters over the transducers.

Parameter	PNP(MPa)	I_SPPA_(W/cm^2^)	I_SPTA_(mW/cm^2^)	MI
Values	0.4144	4.8917	20.0305	0.3869

PNP: peak negative pressure; I_SPPA_: spatial peak pulse average intensity; I_SPTA_: spatial peak temporal average intensity; MI: mechanical index; Test condition: 1.1 MHz, 60 V_pp_, 500 Hz PRF, 10 bursts.

## Data Availability

The data supporting reported results can be obtained by contacting the corresponding author.

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
