# Peer review of "Efficacy Estimation of Microbubble-Assisted Local Sonothrombolysis Using a Catheter with a Series of Miniature Transducers"

_micromachines, 2021, doi:10.3390/mi12060612_

Round 1

Reviewer 1 Report

This article aims to explore some parametric studies for sono-thrombolysis using a side-viewing interstitial ultrasound transducer. The transducer, used in this study, was made of multiple numbers of planar piezoelectric elements, integrated into a catheter structure incorporating a drug-delivering tube. Followed by the transducer characterization, the in vitro test was conducted in a static flow condition. The parametric conditions regarding the clot formation were investigated in terms of calcium ion volume, cold preparation time, and standing time. Meanwhile, the sonothrombolysis effect was studied in terms of microbubble concentration, input voltage, duty cycle, and treatment time. Finally, the sono-thrombolysis was conducted with the chosen parameters combined with urokinase. Materials and approaches look technically reasonable, and the result showed a meaningful clot lysis rate (~70%). However, the current form of the manuscript still leaves some unclear descriptions and discussions in the method and the result. Also, it was not clear what was the merit compared to the existing commercial device (i.e., EKOS). Hence, in my opinion, this paper can be considered for publication by the Micromachines after the major modification. Followings are some additional comments and questions on this paper:

  1. It needs to rewrite the abstract section to clearly describe the research objective and the research significance.

  1. Existing sono-thrombolysis studies mostly utilized rt-PA. Is there any specific reason that the authors employed ‘urokinase’? Please, see “Goldhaber S Z; Kessler C M; Heit J et al. Randomised controlled trial of recombinant tissue plasminogen activator versus urokinase in the treatment of acute pulmonary embolism. Lancet 1988 Aug 6 2 293298”

  1. Specific dimensions of the ultrasound transducer needs to be included in the manuscript; for example, materials and dimension of matching and conductive sheet, and specification of the electric wires.

  1. Was there any potential of wave interruption by placing the wire in the middle of the matching?

  1. In the test setup, what was the microbubble injecting rate (i.e., mL/min)? Besides, the manufacturer of the syringe pump?

  1. Eq. (1) is redundant. When characterizing an acoustic pressure output, most readers already know that the value was computed by the hydrophone sensitivity.

  1. For Eq. (2) and (3), what are PMR and VMR? I guess they should be ALR and AVLR.

  1. On page 3, it quotes, “In the presence of the FEP sleeve, the spatial-peak pressure amplitude was 5% less than the pressure at the transducer surface.” Any technical background for this claim?

  1. On page 5, “from 10 um to 1 um” -> “from 1 um to 10 um”

  1. Detail of the method to obtain the result of section 3.1 needs to be included in section 2.1.

  1. In Table 1, use the superscript for intensity unit, not using “^”. Also, I guess one of the intensities should be ISPPA. I would like to suggest the averaged values for each parameter over the transducers in the table.

  1. For Fig. 8, what is the axial and the lateral direction of the catheter axis?

  1. On page 8, it quotes, “A K-type thermocouple was applied to measure the temperature of thrombus interior …” The purpose of the thermocouple integration needs to be mentioned in the method section (i.e., section 2.2), not in the result section.

  1. The authors claimed that their method could induce inertial and stable cavitation. It quotes, “both stable cavitation and inertial cavitation were likely present in the ultrasound insonation zone.” According to my computation, the mechanical index was below 0.29 (from 0.304 MPa PNA and 1.1 MHz). Is it enough to result in the mechanical index? In the reference [27], Petit et al. considered 200, 350, and 1,300 kPa for the PNP. Where did you get the value of “250kPa”?

  1. The authors need to clearly address the innovation of their study over other past literature. The parametric study has been already conducted in other existing papers as well, in terms of input voltage, duty cycle, treatment time, and MB dose.

  1. Any discussion of the reason why there is no more increase in the lysis rate at the duty cycle over 5%? Similarly, for the voltage level over 60?

  1. Use the high-quality images for the graphs.

Reviewer 2 Report

The manuscript by Huang el al. has described an interesting study. They studied for sonothrombolysis using catheter type US transducer, microbubbles and urokinase. It was used to enhance the thrombolysis effect. The study shows characteristic of blood clot, selection of US parameter, and comparison of treatment results of with/without US, microbubble and urokinase.  The results are interesting and could be a good reference for future studies. However, the idea using microbubble and urokinase was not fresh. Treatment using US catheter for over 30 min could have a challenge to apply to human clinic.

Overall, the development of the catheter type sonothrombolysis with microbubble and urokinase is demonstrating the enhanced clot lysis rates. The study may be required revision showing more advantages of the technique, potential for clinical application, and higher sample numbers.

  1. How many samples are used for statistical results? For results in Figure5, 9, 12 and 13, the sample number should be shown in the figure region. If all data was from three samples, could you increase the sample number at least five?
  2. Figure 6(c) shows -0.2 MPa of negative pressure but (b) shows around 0.4 MPa of PNP at 70 Vpp. Is it correct?
  3. In figure 10, how long time was it conducted and measured the temperature?
  4. In Conclusion, “(Figure 12b)” and “group 7%” should be exchanged.

Round 2

Reviewer 1 Report

The authors addressed all my comments,
yet there exist some minor things to be corrected as follows:

1. In Table 1, use 'x' instead of '*'.

2. The unit of Fig. 8 should be in 'Pa' unit to represent the acoustic pressure field, not squared voltage.

It doesn't need for the revised manuscript to be resent to me for this issue. 

Reviewer 2 Report

None